# Selection of Appropriate Dogs to Be Therapy Dogs Using the C-BARQ

**DOI:** 10.3390/ani13050834

**Published:** 2023-02-24

**Authors:** Mayu Sakurama, Miki Ito, Yumiko Nakanowataru, Takanori Kooriyama

**Affiliations:** Department of Veterinary Science, Rakuno Gakuen University, Ebetsu 069-8501, Hokkaido, Japan

**Keywords:** animal-assisted intervention, C-BARQ, dogs, factor analysis

## Abstract

**Simple Summary:**

Animal-assisted interventions (AAI) help people relax and psychologically recover. Dogs are the most frequently used animals for this purpose and are sometimes called “therapy dogs” in Japan. Certified therapy dogs are evaluated prior to training with an aptitude test, but some owners have their dogs take this test without understanding what is required of the test. Therefore, new methods are needed to recruit candidate dogs and allow owners to easily determine whether their dogs have the potential to be therapy dogs. This study examined which items in Canine Behavior Assessment and Research Questionnaire (C-BARQ) can help to identify dogs suitable to become therapy dogs. Factor analysis identified 14 factors. Using these factors, owners can independently evaluate whether their dogs display suitable behavioural traits to become therapy dogs. The present study may help to increase the number of therapy dogs.

**Abstract:**

In recent years, therapy dogs in medical and assisted living facilities have become popular in Japan, and the demand for such dogs has increased. However, some owners have their dogs take this test, which evaluates the dog’s talent, without understanding what is required of the test. The system needs to teach owners in an understandable way whether their dog is suitable to become a therapy dog so that the owners can determine if their dog is ready to be tested. Therefore, we suggest that easy at-home testing is likely to encourage dog owners to apply for their dog to take the aptitude test. If more dogs take the test, more therapy dogs can be born. The purpose of this study was to identify the personality traits of therapy dogs that pass the aptitude test by using the Canine Behavior Assessment and Research Questionnaire (C-BARQ). The C-BARQ was administered to dogs that previously passed the aptitude test for therapy training at the Hokkaido Volunteer Dog Association, assessing their behavioural displays. A factor analysis was conducted for each questionnaire item, and a total of 98 items were analyzed. Data were collected from the results of 110 dogs encompassing 30 dog breeds, with the most common breeds including Labrador Retrievers, Golden Retrievers, and Toy Poodles. Factor analysis revealed that 14 extracted factors should be evaluated. Given these personality traits and the fact that breed and age did not influence aptitude, we believe that a variety of dogs have the potential to become therapy dogs.

## 1. Introduction

Dogs have long been familiar to humans and are animals that can provide psychological healing through their interactions with people [1]. In recent years, this healing effect has been scientifically demonstrated in several animal species, and awareness of animal-assisted intervention (AAI) has increased [1]. Interactions with dogs or other animal species are used in medical and assisted living facilities (i.e., hospitals and nursing homes) to improve people’s quality of life (QOL) [1]. A variety of animals, from horses to dolphins, have been used in animal therapy, but dogs have become mainstream in recent years [2,3,4,5]. Specific effects of animal therapy have been reported, including decreased progression of depression [2,6,7], stabilization of blood pressure [8,9], and increased motivation to socialize [10,11,12]. In expectation of these effects, many hospitals and facilities have introduced dog visits, and the demand for these services has been increasing.

AAI can be divided into three categories according to its methods. The first is animal-assisted therapy (AAT). This is a type of subsequent therapy led by medical professionals. The second is animal-assisted education (AAE). This refers to activities that involve animals in educational fields. The third is animal-assisted activity (AAA), which involves activities that do not have a set therapeutic purpose and are mainly focused on interacting with dogs with the aim of improving the QOL of the subjects [13]. Currently, AAA is the most common activity in Japan, and the dogs involved are generally called “therapy dogs” there. In Japan, there is no legal certification for a therapy dog, but dogs and handlers with a “therapy dog certification” are active in this field. To become certified, dogs must pass the Therapy Dog Aptitude Test. In this test, conducted by the Hokkaido Volunteer Dog Association, candidates are evaluated on the following content: gentleness of the dog’s reactions to other dogs, a person touching the dog’s body, loud sounds, and appearance of wheelchairs and obedience to the sit command and to walk at their owner’s side. If the dog barks, engages in any dangerous behavior, urinates or defecates during the test, the test is terminated immediately [14]. Opportunities for this test are limited to twice a year, and the aptitude test seems to be difficult for most dogs. Therefore, it is challenging to recruit a large number of candidate therapy dogs. To provide a wider selection of therapy dogs, a new selection system is required that encourages owners to have their dogs take an aptitude test. This study presents an instant test that can be conducted before the aptitude test to allow dog owners to judge at home whether their dogs have the potential to be therapy dogs.

There are some tests that judge dogs’ personality by questionnaires. The Canine Behaviour Assessment and Research Questionnaire (C-BARQ) is a well-known dog personality test that allows owners to evaluate their dogs. The C-BARQ is also applied for therapy dog assessment in other countries [15,16]. We suggest that the C-BARQ could be a useful tool for identifying dogs with suitable characteristics to be therapy dogs.

Based on the above background, the purpose of this study is to assess whether the C-BARQ could be used at home by owners to see whether their pet dog has the right characteristics to pass the aptitude test. In this study, we extracted the personality factors of certified therapy dogs through a questionnaire survey using the C-BARQ.

## 2. Materials and Methods

### 2.1. Dogs

Dogs that passed the Hokkaido Volunteer Dog Association’s therapy dog aptitude test and were currently or had previously been involved in therapy dog activities were surveyed with regard to their behaviour patterns under various circumstances. The dogs included were active dogs, retired dogs, and deceased dogs. For deceased dogs, owners were asked to recall the dog’s behavior when answering the questionnaire. This aptitude test assesses only the dog’s aptitude, not that of the owners. Dogs that pass the aptitude test must be retested every two years, because their temperament can change. Contact information of people with dogs that had passed the aptitude test was obtained with the help of the Hokkaido Volunteer Dog Association. The Hokkaido Volunteer Dog Association is a non-profit organization (NPO) established in 1996 that visits various places, including medical institutions and assisted living facilities (i.e., hospitals and nursing homes) [14].

### 2.2. Questionnaire

The C-BARQ was used for the survey (version released in 2020). It was developed by the University of Pennsylvania researchers Serpell et al. in 2003 and has been used in many behavior analysis studies as a diagnostic index of canine personality [17]. The questionnaire consists of 101 questions divided into 13 categories: stranger-directed aggression, owner-directed aggression, dog-directed aggression/fear, trainability, chasing, stranger-directed fear, nonsocial fear, dog-directed fear, separation-related behavior, touch sensitivity, excitability, attachment/attention-seeking, and energy level. Surveys were conducted twice, once in July 2020 via direct e-mail to 112 dog owners, and again in October 2021 via direct e-mail to 115 dog owners. The responding owners rated and answered the C-BARQ questions on a five-point scale from 0 to 4. Other basic information was gathered including the dog’s name, date of birth, and breed. This study was conducted with the approval of the Ethical Regulations for Animal Experiments of Rakuno Gakuen University (VH19B9).

### 2.3. Factor Analysis Methods

After compiling the collected data into a spreadsheet, a factor analysis was conducted on the questionnaire items. The statistical software R ver. 3.5.3. (The R Foundation, Vienna) was used for factor analysis. Samples with a response rate of at least 75% to the survey questions and questions with an overall response rate of at least 85% were used in the factor analysis [18]. The samples obtained from these criteria were used to determine the number of factors based on the MAP (Minimum Average Partial) criteria. After estimating the factor loadings by the minimum residual method, oblique rotation of the axes was performed by the oblimin method. Items with factor loadings of 0.5 or higher and commonality of 0.3 or higher were selected, and the factors obtained were named.

## 3. Results

In this study, therapy dog aptitude factors were extracted from dogs that passed the therapy dog aptitude test using the C-BARQ questionnaire.

### 3.1. Questionnaire Collection Rate

A total of 114 questionnaires were collected as a result of the aggregated first- and second-round questionnaire data. Four questionnaires with insufficient data were excluded; thus, data from 110 dogs were used for factor analysis. The collection rate was 59.7%. Of these, 63 were dogs from multi-dog homes.

### 3.2. Dog Breed and Age Distribution

The three most common breeds of therapy dogs in the Hokkaido Volunteer Dog Association were represented: the Labrador Retriever (LR), Golden Retriever (GR), and Toy Poodle (TP). Other breeds included mixed breed, Shih Tzu, Miniature Dachshund, Cavalier King Charles Spaniel, Yorkshire Terrier, Samoyed, Beagle, Miniature Poodle, Miniature Schnauzer, Shetland Sheepdog, Border Collie, Shiba Inu, Maltese, Papillon, Newfoundland, Standard Poodle, Boston Terrier, Australian Labradoodle, Pembroke Welsh Corgi, Pomeranian, Chihuahua, Bernese Mountain Dog, Australian Shepherd, Pug, Löwchen, Goldendoodle, and Brussel Griffon, for a total of 27 breeds in addition to the three main ones. Regarding age, the active dogs were between 2 and 15 years old, with a mean age of 8.53 years (±2.82) (Figure 1). There were 9 retired dogs and 27 deceased dogs, which were not included in the age distribution.

### 3.3. Results of Factor Analysis

In total, data for 110 animals and 98 questions were used for factor analysis. Factor analysis was performed on the 98 questions, and 51 items were selected (Table 1). These 51 items were grouped into 14 factors containing similar items, and named according to the included items. Two of the factors were obtained only from dogs in multi-dog households. The 14 extracted factors (Table 1) were as follows: trainability and obedience, separation-related anxiety behaviors, separation-related physiological reactions, aggression toward people, fear and anxiety toward strangers, fear and anxiety toward dogs, fear and anxiety in unfamiliar situations, territorial aggression, hyperactivity, aggression toward approaching unknown dogs, resource guarding-related aggression, abnormal behavior, excitability, attachment to family members, vigilant aggression toward family dogs, and resource guarding-related aggression toward family dogs.

The results of the average scores (Table 2) indicated that therapy dogs tended to score highly on factors of trainability and obedience, low to moderate on attachment behavior to family members, and low on excitability. In contrast, therapy dogs tended to have low scores on the following factors: separation-related anxiety behaviors, separation-related physiological reactions, aggression toward people, fear and anxiety toward strangers, fear and anxiety toward dogs, fear and anxiety in unfamiliar situations, territorial aggression, hyperactivity, aggression toward approaching unknown dogs, resource guarding-related aggression, abnormal behavior, vigilant aggression toward family dogs, and resource guarding-related aggression toward family dogs.

## 4. Discussion

### 4.1. Breed, Age, and Housing Environment

The fact that dogs of various breeds and a wide range of ages are used as therapy dogs suggests that dogs of many age groups may have aptitude for the role. Regarding breeds, the reason the LR, GR, and TP are the top three might be related to each dog breed’s personality and popularity. LRs and GRs show low aggression toward people and dogs, and high trainability [19,20,21]. The TP is the most popular dog breed in Japan [22] and is considered “faithful and trainable” in terms of personality, which may have helped place it at the top of the list. Regarding the age distribution of the therapy dogs, the mean age was higher than that of general house dogs [23]. The age distribution was also higher than that of general dogs, which may be due to the lack of new therapy dogs, especially the lack of young individuals. Additionally, 55.3% of the therapy dogs were kept in multi-dog households. Kubiny et al. [24] reported that “more experience with prior dogs is associated with higher calmness”, which suggests dogs in multi-dog households may have suitable temperaments for animal-assisted intervention.

### 4.2. Extracted Factors and Their Relevance to the Therapy Dog Temperament Test

The factors were divided into two groups: one was included in the aptitude testing, and the other was not included in the test.

#### 4.2.1. Factor Content Tested in the Aptitude Test

For high “trainability and obedience”, dogs must obey their owner’s command to “sit” and walk the course with the owner properly. If the dog’s training and obedience are at a low level, the dog may act selfishly due to a lack of control, and therapy activity cannot be carried out smoothly. Therapy animals are required to be obedient to their handler’s commands while paying constant attention to them [25]. For low levels of “aggression toward people” and “fear and anxiety toward strangers/dogs/unfamiliar situations”, the dogs should not show aggression or fear/anxiety toward “any dogs, ordinary people or suspicious people”, “approaching any dogs, ordinary people or suspicious people” or “a person who touches any part of their body” in an unfamiliar place (the testing room). Touching by residents is an essential part of AAI activity in care facilities, but sometimes the residents grasp the dog firmly. The dogs were evaluated for their reaction to simulated residents, especially those in wheelchairs or with canes. In addition, the locations of activities vary, and unfamiliar situations elicit fear and anxiety in dogs. If these factor scores are high, the dog is not suitable as a therapy dog. In the worst-case scenario, the dog will hurt people as a defensive behavior during the activity. Therapy dogs are required to have a temperament that allows them to “remain calm and gentle, and prefer to stay close by people”, be “adaptable to unfamiliar situations such as novel scenes, sounds, or smells”, “accept unfamiliar persons without fear”, “ignore neutral dogs” and “never show aggressive behavior” [25].

#### 4.2.2. Factors Not Included in the Aptitude Test

With regard to low levels of “separation-related anxiety behaviors” and “separation-related physiological reactions”, dogs with separation anxiety and separation-related physiological reactions may have stressful feelings in their ordinary life and may feel anxiety in unfamiliar situations. Gerrard et al. [26] noted that separation anxiety is significantly related to excessive attachment to owners. Neither attachment to household members nor separation anxiety should be high in therapy dogs. In terms of low levels of “territorial aggression”, dogs that show territorial aggression may show aggression toward other dogs or people who approach them. For low levels of “hyperactivity”, if a dog cannot control itself and reacts excessively, it may not be able to pay proper attention to the owner’s commands. If the dog has high hyperactivity, people may feel uncomfortable and may be injured. It is necessary for therapy dogs to regain control even after playing or excitement [25]. Low levels of “resource guarding-related aggression” are also necessary. A dog that exhibits aggressive behavior to avoid having food or other necessary items taken away is highly unlikely to be suitable as a therapy dog, since the dog may have a strong tendency toward anxiety [27]. Low levels of “aggression toward approaching unknown dogs” are also necessary; therapy dogs should not show aggression regardless of the proximity of other dogs. With regard to low levels of “abnormal behavior”, compulsive behavior and instinctual drive disorder may disturb peaceful contact with people in a care home. Low levels of “vigilant aggression toward family dogs” and “resource guarding-related aggression toward family dogs” are necessary; among the therapy dogs in the present study, more than half (55.3%) were in multi-dog households.

Mild “excitability” is necessary for therapy dogs. Dogs show excitement in response to playing, walking outside, food rewards, and hunting instinct stimuli, as well as owners returning home and car rides. However, dogs with high excitability behave intensely and are at risk of injuring people in care homes. It is necessary for therapy dogs to retain control even during excitable moments [25]. With regard to mild to moderate “attachment behavior to family members”, high-attachment dogs, such as those that sleep with their owners, tend to show separation anxiety and aggression [26,28], or tend to show attention-seeking behavior and separation anxiety [29]. If the degree of attachment of a pet dog to its owner is low, “aggressive behavior” is often observed [30]. This suggests that a mild to moderate level of attachment is necessary because attachment that is too high leads to behaviors such as barking, scratching and urination.

These extracted factors are the aspects of temperament that owners should check to determine whether their dogs may be candidate to act as therapy dogs. Therefore, if the C-BARQ could be placed on the website of the Hokkaido Volunteer Dog Association, owners could independently evaluate whether their dogs have suitable temperaments to be therapy dogs. This might lead to an increase in the number of dogs that take the aptitude test. Furthermore, the results showed that low aggression is important for therapy dogs. However, some owners are not aware that a lack of aggression is a prerequisite for becoming a therapy dog [31]. Therefore, the C-BARQ may help owners to understand the requirements of therapy dogs and to train their dogs to pass the therapy dog aptitude test. In addition, only some of the factors are included in the therapy dog aptitude test. Thus, the current test method does not fully evaluate therapy dog aptitude. Therefore, the C-BARQ may also be used as an initial screening instrument for the therapy dog aptitude test to better select dogs suitable for becoming therapy dogs. Furthermore, ordinary dog owners can complete the entire C-BARQ, which includes these questions. Uninterested owners can be directed to AAA. Additionally, the expression of temperament (i.e., behavioral displays) can change over time, despite the general perception that temperament is genetic and thus remains consistent over time, activities and environment. Therefore, therapy dogs must renew their certification every two years. These factors must be interpreted for future therapy dogs as well as active dogs. Since only dogs that passed the therapy dog aptitude test were included in this study, future studies should compare their C-BARQ scores with those of dogs that failed the therapy dog aptitude test to obtain more substantial results.

## 5. Conclusions

As part of our efforts to promote therapy dogs, we used the C-BARQ to identify the characteristics of therapy dogs. We extracted factors that appear to indicate which dogs have the potential to become therapy dogs. Based on these factors, we believe that the C-BARQ will contribute to increasing the number of dog owners who take therapy dog aptitude tests by allowing owners to independently evaluate whether their dogs have suitable temperaments to become therapy dogs. The C-BARQ may also help owners to understand the requirements of therapy dogs and to train their dogs to pass the therapy dog aptitude test.

## Figures and Tables

**Figure 1 animals-13-00834-f001:**
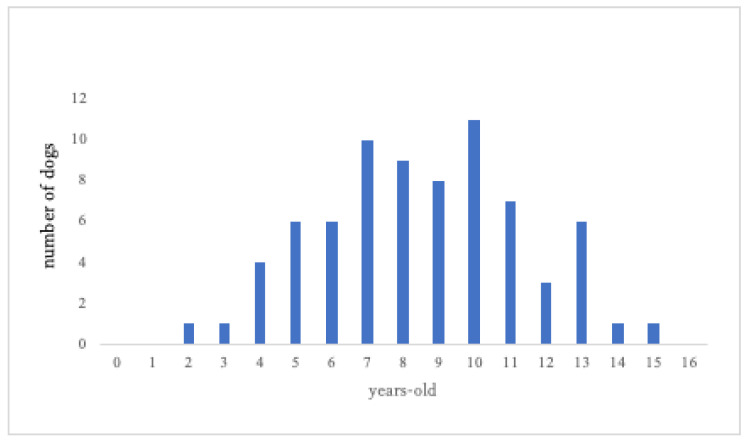
Age distribution of 74 active dogs.

**Table 1 animals-13-00834-t001:** Named factors of therapy dog talent and their original questions with factor scores.

Factors and Questionnaires	Score Mean ± SD	Factor Loadings	Factor Contribution	Factor Contribution Rate	Cumulative Contribution Rate	Cronbach’s α
1. Trainability and obedience	2.7 (±0.91)		2.052	0.256	—	0.6
Obeys a stay command immediately.	3.4 (±0.65)	0.68				
When off the leash, returns immediately when called.	3.2 (±0.81)	0.595				
Slow to learn new tricks or tasks.	1.4 (±0.81)	−0.584				
2. Separation-related anxiety behaviours	0.4 (±0.11)		1.89	0.24	0.24	0.71
Restlessness/agitation/pacing.	0.5 (±1.02)	0.71				
Whining.	0.4 (±0.86)	0.76				
Barking.	0.3 (±0.67)	0.6				
3. Separation-related physiological reactions	0.1 (±0.04)		1.76	0.22	0.46	0.75
Excessive salivation.	0.0 (±0.21)	0.81				
Chewing/scratching at doors, floor, windows, curtains, etc.	0.1 (±0.40)	0.54				
Loss of appetite	0.1 (±0.47)	0.88				
4. Aggression towards people	0.1 (±0.03)		2.768	0.12	0.482	0.8
When stepped over by a member of the household.	0.0 (±0.16)	0.804				
When approached directly by an unfamiliar child while being walked or exercised on a leash.	0.1 (±0.30)	0.638				
When stared at directly by a member of the household.	0.0 (±0.13)	0.623				
When approached directly by an unfamiliar adult while being walked or exercised on a leash.	0.1 (±0.28)	0.524				
When an unfamiliar person tries to touch or pet the dog.	0.1 (±0.26)	0.511				
5. Fear and anxiety towards strangers, dogs, unfamiliar situations	0.6 (±0.29)		5.74	0.3	—	0.87
When approached directly by an unfamiliar male adult while away from the home.	0.2 (±0.55)	0.73				
When approached directly by an unfamiliar child while away from the home.	0.3 (±0.57)	0.8				
When an unfamiliar person tries to touch or pet the dog.	0.2 (±0.46)	0.7				
When approached directly by an unfamiliar dog of the same or larger size.	0.6 (±0.77)	0.75				
When approached directly by an unfamiliar dog of a smaller size.	0.4 (±0.72)	0.67				
When first exposed to unfamiliar situations (e.g., first car trip, first time in elevator, first visit to veterinarian, etc.).	0.9 (±1.03)	0.76				
In response to wind or wind-blown objects.	0.5 (±0.94)	0.65				
When having nails clipped by a household member.	0.9 (±1.11)	0.51				
When barked, growled, or lunged at by an unfamiliar dog.	1.0 (±1.12)	0.62				
6. Territorial aggression	0.3 (±0.11)		5.28	0.23	0.23	0.91
When strangers walk past the home while the dog is in the yard.	0.3 (±0.65)	0.923				
When mailmen or other delivery workers approach the home.	0.4 (±0.70)	0.905				
Toward unfamiliar persons visiting your home.	0.3 (±0.59)	0.868				
When joggers, cyclists, roller skaters, or skateboarders pass the home while the dog is in the yard.	0.3 (±0.57)	0.857				
When an unfamiliar person approaches the owner or a member of the owner’s family at home.	0.2 (±0.49)	0.712				
Toward cats, squirrels, and other animals entering its yard.	0.5 (±0.76)	0.507				
7. Hyperactivity	0.8 (±0.19)		2.948	0.123	0.123	0.86
Active, energetic, always on the go.	0.9 (±0.98)	0.805				
Playful, puppyish, boisterous.	1.0 (±1.08)	0.766				
Hyperactive, restless, has trouble settling down.	0.5 (±0.86)	0.707				
8. Aggression towards approaching unknown dogs	0.4 (±0.0)		3.036	0.132	0.362	0.99
When approached directly by an unfamiliar male dog while being walked or exercised on a leash.	0.4 (±0.71)	0.953				
When approached directly by an unfamiliar female dog while being walked or exercised on a leash.	0.4 (±0.73)	0.951				
9. Resource guarding-related aggression	0.1 (±0.44)		2.274	0.099	0.581	—
When food is taken away by a member of the household.	0.1 (±0.44)	0.618				
10. Abnormal behavior	0.6 (±0.02)		2.868	0.12	0.242	0.5
Licks people or objects excessively.	0.6 (±0.84)	0.619				
Rolls in animal droppings or other “smelly” substances.	0.5 (±0.89)	0.554				
11. Excitability	1.5 (±0.09)		3.226	0.538	—	0.87
When visitors arrive at its home.	1.6 (±1.14)	0.838				
Just before being taken for a walk.	1.5 (±1.13)	0.793				
Just before being taken on a car trip.	1.6 (±1.18)	0.78				
When you or other members of the household come home after a brief absence.	1.4 (±1.14)	0.713				
When playing with you or other members of your household.	1.4 (±1.01)	0.637				
When the doorbell rings.	1.6 (±1.36)	0.61				
12. Attachment behaviour to family members	1.8 (±0.49)		2.629	0.438	—	0.8
Tends to sit close to or in contact with a member of the household when that individual is sitting down.	2.4 (±1.21)	0.83				
Tends to nudge, nuzzle, or paw a member of the household for attention when that individual is sitting down.	1.7 (±1.19)	0.733				
Becomes agitated when a member of the household shows affection for another dog or animal.	1.0 (±1.26)	0.676				
Tends to follow a member of household from room to room about the house.	1.9 (±1.26)	0.636				
13. Vigilant aggression towards family dogs	0.2 (±0.01)		1.689	1.69	0.42	0.89
When approached at a favorite resting/sleeping place by another (familiar) household dog.	0.2 (±0.70)	1.015				
Towards another (familiar) dog in your household.	0.3 (±0.64)	0.773				
14. Resource guarding-related aggression towards family dogs	0.4 (±0.07)		1.340	1.34	0.76	0.78
When approached while playing with/chewing a favorite toy, bone, object, etc., by another (familiar) household dog.	0.5 (±0.73)	0.914				
When approached while eating by another (familiar) household dog.	0.3 (±0.64)	0.67				

Score Mean ± SD: The response scores (0–4) for each factor and question are averaged, and SD indicates standard deviation. Factor Loadings: Value indicating the extent to which each question is reflected in the factor. Factor Contribution: The square of the factor loading indicates the amount of variance in the observed variable that the factor can explain. Factor Contribution Rate: Factor contribution divided by number of items. Cumulative Contribution Rate: Cumulative total of factor contributions. Cronbach’s α: reliability factor.

**Table 2 animals-13-00834-t002:** Trends in each factor of therapy dog temperament traits.

Factor	Trend
Trainability and obedience	High
Attachment behaviour to family members	Mild to moderate
Excitability	Mild
Separation-related anxiety behaviours	Low
Separation-related physiological reactions	Low
Aggression towards people	Low
Fear and anxiety towards strangers, dogs, unfamiliar situations	Low
Territorial aggression	Low
Hyperactivity	Low
Aggression towards approaching unknown dogs	Low
Resource guarding-related aggression	Low
Abnormal behaviour	Low
Vigilant aggression towards family dogs	Low
Resource guarding-related aggression towards family dogs	Low

## Data Availability

The data presented in this study are not publicly available. Please contact the corresponding author with any enquiries.

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
