# Peer review of "Selection of Appropriate Dogs to Be Therapy Dogs Using the C-BARQ"

_animals, 2023, doi:10.3390/ani13050834_

Round 1
Reviewer 1 Report
Summary
The paper used the C-BARQ questionnaire to identify potential important factors for therapy dogs. Fourteen factors were identified during the analysis from a range of breeds. Authors suggest pet owners could use these factors to self-evaluate whether their dog has a suitable temperament to be therapy dogs. The paper will be of interest to pet dog owners and those working in the therapy and assistant dog sectors.
General comments
The area of therapy dog selection is of increasing scientific interest and has valuable practical applications. The paper is well written, and the structure is generally easy to follow. However, the study could be enhanced by including a comparison of CBARQ scores for dogs which are not considered suitable to be therapy dogs or population averages. Including the potential limitations of the approach to therapy dog selection (e.g. owner biases, mis-interpretations etc) as well as benefits would enhance the discussion. In addition, a more indepth disucssion on the application of the presented approach in the therapy dog selection process would be of value in the discussion (ie in addition to, potential alternative to the apptitude test, inital screener etc.). This could also include an expansion of the implications of the results on the availability of potential therapy dogs.
Line 239: Suggest edit to say that the expression of temperament can change (ie behavioural displays) as temperament is generally accepted to be genetic and so remains consistent over time, activities and environment
Author Response
Dear Reviewer 1
>Thank you very much for your comments. I hope the revised manuscript meets your comments.
General comments
The area of therapy dog selection is of increasing scientific interest and has valuable practical applications. The paper is well written, and the structure is generally easy to follow. However, the study could be enhanced by including a comparison of CBARQ scores for dogs which are not considered suitable to be therapy dogs or population averages. Including the potential limitations of the approach to therapy dog selection (e.g. owner biases, mis-interpretations etc.) as well as benefits would enhance the discussion. In addition, a more in-depth discussion on the application of the presented approach in the therapy dog selection process would be of value in the discussion (i.e. in addition to, potential alternative to the aptitude test, initial screener etc.). This could also include an expansion of the implications of the results on the availability of potential therapy dogs.
>Yes. I agree with your comments about using the C-BARQ for therapy dog selection. The true purpose of this study is to enhancing the animal assisted intervention in Japan. And we also agree to add more discussion about the C-BARQ prior to the aptitude test which encourage the dog owners to take the aptitude test.
>The result of the present study should be compared with the aptitude test rejected applicant. So, we added the limitation in the discussion. We are going to collect the C-BARQ score of rejected applicant in the next approach.
Line 239: Suggest edit to say that the expression of temperament can change (i.e. behavioural displays) as temperament is generally accepted to be genetic and so remains consistent over time, activities and environment
>Thank you for your advice. decided to change “temperament” to “personality” or “behavioural display”
Reviewer 2 Report
Thank you for submitting this interesting paper to Animals.
The title of the article is unclear. Maybe it should be “The use of the C-BARQ to select dogs appropriate to be therapy dogs” or something similar.
It is unclear in several places about whether the aptitude test is for the dogs, the owners or both. It appears that it is really an aptitude test for the dogs (see line 80) so please do not use it in relation to the owners. Your study assessed whether the C-BARQ could be used at home by owners to see whether their pet dog had the right characteristics to pass the aptitude test. This seems like a good idea.
In several places you talk of “the medical and welfare fields’ (lines 19, 39). It is unclear what you mean by welfare settings or fields. Please clarify or change terminology.
Simple summary: Please clarify that it is the available times for the taking of the test that is the bottle neck (not the test itself).
The use of ‘important factors’ is too vague (line 15). Maybe: ‘This study examined which items in C-BARQ would be helpful in identifying dogs suitable to be trained therapy dogs. Fourteen items …’
Abstract: Again reword about bottleneck. Also, dog owners cannot apply to be a therapy dog. Reword line 22.
Line 27 – 110 dogs were not collected. Rather data (or results) from 110 dogs were collected.
Introduction: First sentence is awkward. Many of the references used in the introduction are old and from book chapters. Please find more up-to-date references in the literature.
The division of AAI is important and your description is useful. However, we need a reference for the division (7 at the end is not good enough). It may be your division which is OK but make that clear. Also, it would be best to acknowledge that the terminology around assistance animals is currently problematic but you could then say this is how you are classifying it in Japan.
Please be careful in your use of commas (e.g. unnecessary commas in lines 65 & 66).
Please reword the purpose of the study as discussed above.
Method: Please clarify how you obtained the contact details of people who owned dogs that had passed the aptitude test. Had they had to have passed within a particular period and/or worked as therapy dogs in the last two years for example? Later you mention that certification must be repeated every two years – this should ne mentioned here.
Results: What does a group feeding environment mean? (line 116)
Line 118-9 says three breeds were represented but then lists many breeds (20). Please clarify. What does it mean to say therapy dogs had a higher mean age? Higher than what? And active dogs tended to be older? Please clarify these two statements. Please also clarify the numbers. 114 surveys were collected but 9 were for retired and 27 for deceased dogs which leaves 78. But the graph Figure 1 only has 74 active dogs ages includes. Please explain. Then in 3.3 you say you used data from 110 animals – explain.
Please clarify the difference between the 51 items and then the 14 factors.
Discussion: TP are the most popular dog – is this true world wide of just in Japan?
Discussion and conclusion: Please make it much clearer what the benefits of your findings are and how it will help overcome the bottleneck you mentioned earlier.
Author Response
Dear Reviewer 2
>Thank you very much for your comments. I hope the revised manuscript meets your comments.
The title of the article is unclear. Maybe it should be “The use of the C-BARQ to select dogs appropriate to be therapy dogs” or something similar.
> We agreed to change the title, and then we introduced “Selection of appropriate dogs for “Therapy dog” with the C-BARQ”.
It is unclear in several places about whether the aptitude test is for the dogs, the owners or both. It appears that it is really an aptitude test for the dogs (see line 80) so please do not use it in relation to the owners. Your study assessed whether the C-BARQ could be used at home by owners to see whether their pet dog had the right characteristics to pass the aptitude test. This seems like a good idea.
>We changed the sentence not using owners but dogs on Line 81
> We kindly referred you last sentence on Line 74-75
In several places you talk of “the medical and welfare fields’ (lines 19, 39). It is unclear what you mean by welfare settings or fields. Please clarify or change terminology.
>We changed those words to “medical and assisted living facilities”.
Simple summary: Please clarify that it is the available times for the taking of the test that is the bottle neck (not the test itself).
>We changed the bottle neck to a hurdle in this whole manuscript. The sentence was changed to “but this test poses a hurdle for the dog owners who are interested in AAI” on Line 9-10.
The use of ‘important factors’ is too vague (line 15). Maybe: ‘This study examined which items in C-BARQ would be helpful in identifying dogs suitable to be trained therapy dogs. Fourteen items …’
>We kindly referred to your sentence and modified like this “This study examined which items in Canine Behavior Assessment and Research Questionnaire (C-BARQ) can help to identify dogs suitable to become therapy dogs” on Line 12-13
Abstract: Again reword about bottleneck. Also, dog owners cannot apply to be a therapy dog. Reword line 22.
> Yes. We did it, too.
Line 27 – 110 dogs were not collected. Rather data (or results) from 110 dogs were collected.
>We modified the sentence as “Data were collected from results of 110 dogs encompassing 30 dog breeds” on Line 27-38. Is this OK?
>We also modified the sentence on Line 122-123 in Results as “Four questionnaires with insufficient data were excluded; thus, data from 110 dogs were used for factor analysis”
Introduction: First sentence is awkward. Many of the references used in the introduction are old and from book chapters. Please find more up-to-date references in the literature.
>Thank you for your comments. Yes. We changed them.
The division of AAI is important and your description is useful. However, we need a reference for the division (7 at the end is not good enough). It may be your division which is OK but make that clear. Also, it would be best to acknowledge that the terminology around assistance animals is currently problematic but you could then say this is how you are classifying it in Japan.
>We added “in Japan” in the sentence on Line 52-53 and also other parts, because this word “Therapy dogs” are very common in Japan.
Please be careful in your use of commas (e.g. unnecessary commas in lines 65 & 66).
>Thank you for your advice. We corrected them.
Please reword the purpose of the study as discussed above.
> Yes. We kindly referred you last sentence on Line 74-75.
Method: Please clarify how you obtained the contact details of people who owned dogs that had passed the aptitude test. Had they had to have passed within a particular period and/or worked as therapy dogs in the last two years for example? Later you mention that certification must be repeated every two years – this should ne mentioned here.
>We modified the sentences as “The dogs included were active dogs, retired dogs, and deceased dogs. For deceased dogs, owners were asked to recall the dog’s behavior when answering the questionnaire. This aptitude test assesses only the dog's aptitude, not that of the owners. Dogs that pass the aptitude test must be retested every two years, because their temperament can change. Contact information of people with dogs that had passed the aptitude test was obtained with the help of the Hokkaido Volunteer Dog Association” on Line 83-89 referring to your advices. We hope these sentences meted to your comments.
Results: What does a group feeding environment mean? (line 116)
>This sentence is meaning multi dog rearing owners’ home. So, we modified as “multi-dog homes”
Line 118-9 says three breeds were represented but then lists many breeds (20). Please clarify. What does it mean to say therapy dogs had a higher mean age? Higher than what? And active dogs tended to be older? Please clarify these two statements.
>We discussed with members and these sentences were unnecessary and might induce confusion. So, we removed those sentences.
Please also clarify the numbers. 114 surveys were collected but 9 were for retired and 27 for deceased dogs which leaves 78. But the graph Figure 1 only has 74 active dogs ages includes. Please explain. Then in 3.3 you say you used data from 110 animals – explain.
>The number that used this analysis was 110 because 4 questionnaire answers were invalid. So, 110 – (9 retired + 27 diseased + 74 active). We apologize the confusions.
Please clarify the difference between the 51 items and then the 14 factors.
>We modified the sentence as “These 51 items were grouped into 14 factors containing similar items and named according to the included items” in Line 140-142.
Discussion: TP are the most popular dog – is this true world wide of just in Japan?
>We added “in Japan”, because TP is most popular breed in Japan.
Discussion and conclusion: Please make it much clearer what the benefits of your findings are and how it will help overcome the bottleneck you mentioned earlier.
>Yes. We referred your whole comments and add some sentences. We hope those sentences met your comments.
>We modified the sentences on Line 242-262 in Discussion and on Line 267-270 Conclusions as:
Discussion:
“Therefore, if the C-BARQ could be placed on the website of the Hokkaido Volunteer Dog Association, owners could independently evaluate whether their dogs have suitable temperaments to be therapy dogs. This might lead to increases in the number of dogs that take the aptitude test. Furthermore, the results showed that low aggression is important for therapy dogs. However, some owners are not aware that a lack of aggression is a prerequisite for becoming a therapy dog [15]. Therefore, the C-BARQ may help owners to understand the requirements of therapy dogs and to train their dogs to pass the therapy dog aptitude test. In addition, only some of the factors are included in the therapy dog aptitude test. Thus, the current test method does not fully evaluate therapy dog aptitude. Therefore, the C-BARQ may also be used as an initial screening instrument for the therapy dog aptitude test to better select dogs suitable to become therapy dogs. Furthermore, ordinary dog owners can complete the entire C-BARQ, which includes these questions. Uninterested owners can be directed to AAA. Additionally, the expression of temperament (i.e., behavioral displays) can change over time, despite the general perception that temperament is genetic and thus remains consistent over time, activities and environment. Therefore, therapy dogs must renew their certification every two years. These factors must be interpreted for future therapy dogs as well as active dogs. Since only dogs that passed the therapy dog aptitude test were included in this study, future studies should compare their C-BARQ scores with those of dogs that failed the therapy dog aptitude test to obtain more substantial results.”
Conclusions:
“by allowing owners to independently evaluate whether their dogs have suitable temperaments to become therapy dogs. The C-BARQ may also help owners to understand the requirements of therapy dogs and to train their dogs to pass the therapy dog aptitude test.”
Round 2
Reviewer 2 Report
Thank you for carefully addressing my previous comments. The paper is not yet quite ready for publication. See comments below.
Title: Is still not quite right. I suggest “Selection of appropriate dogs to be therapy dogs using the C-BARQ”
Simple summary and abstract. The use of the word ‘hurdle’ in the simple summary is still not sufficiently clear. It is not really a hurdle for the dog owners. Perhaps it is a hurdle for the dogs in the sense that many struggle to pass. Perhaps what you are saying is “many owners insufficiently understand the requirements of the aptitude test and put their dogs for assessment when they are not suitable.”
Similarly in the abstract the wording will need to change. The expression “owners to challenge” makes no sense.
Introduction: Lines 47-55. Initially Therapy dogs are describes as AAT but then later as AAA. This must be clarified.
Line 57 please delete that owners have to pass the aptitude test. In line 63 – why is the aptitude difficult for most owners when later you say the test is only for the dogs. Again line 66 says owners take the test.
Line 72 – please reword the last sentence in this paragraph. Maybe “we suggest that the C-BARQ could be a useful tool for identifying dogs with suitable characteristics to be therapy dogs.
Materials and methods
Line 95 please put reference in brackets
Line 104 – reword to “Other basic information was gathered including the dog’s name, date of birth and breed”
Line 110 &111 – Please clarify why you have 75% and 85% and what happened with these answers.
Line 113 – maybe I missed it but what is MAP criteria
Results – Was there a cut off period? I mean, I know you enrolled owners of deceased dogs, but was there a time limit?
Line 139 – please explain why 98 out of a possible 101 were chosen.
Discussion:
Lines 193-4 and 224 appear to say similar things – low levels of aggression towards other dogs.
Conclusion: Line 265 - you extracted factors that appear to indicate that dogs have the potential to become therapy dogs.
Author Response
Dear Reviewer 2
Thank you very much for your comments and advices. We modified the sentences according to your comments and also answered your questions. We hope that our response meets your reviews. The modified parts were highlighted.
Response to Reviewer 2’s comments
Title: Is still not quite right. I suggest “Selection of appropriate dogs to be therapy dogs using the C-BARQ”
> We corrected the title as your advice.
Simple summary and abstract. The use of the word ‘hurdle’ in the simple summary is still not sufficiently clear. It is not really a hurdle for the dog owners. Perhaps it is a hurdle for the dogs in the sense that many struggle to pass. Perhaps what you are saying is “many owners insufficiently understand the requirements of the aptitude test and put their dogs for assessment when they are not suitable.”
> We corrected it with your advice on Line 12 and on Line 20-21.
Similarly in the abstract the wording will need to change. The expression “owners to challenge” makes no sense.
> We modified the sentence on Line 22-23.
Introduction: Lines 47-55. Initially Therapy dogs are describes as AAT but then later as AAA. This must be clarified.
> In Japan, the term "therapy dogs" is used not only for AAT but also AAA. So, we would like to retain this term. Therefore, we have removed the description of AAT's therapy dogs and added "in Japan" to AAA's therapy dogs on Line 54-56.
Line 57 please delete that owners have to pass the aptitude test. In line 63 – why is the aptitude difficult for most owners when later you say the test is only for the dogs. Again line 66 says owners take the test.
> The sentence on Line 57 was corrected. Lines 63 and 66 were rewritten as the dog taking the test.
Line 72 – please reword the last sentence in this paragraph. Maybe “we suggest that the C-BARQ could be a useful tool for identifying dogs with suitable characteristics to be therapy dogs.
> We rephrased it as on Line 73-74.
Materials and methods
Line 95 please put reference in brackets
> We corrected it.
Line 104 – reword to “Other basic information was gathered including the dog’s name, date of birth and breed”
> We modified it as on Line 104-105.
Line 110 &111 – Please clarify why you have 75% and 85% and what happened with these answers.
>We analyzed our data according to Reference 19 in Materials and Methods. We adopted 75% because it is generally accepted. The 85% covers more strict standards than 75%.
Line 113 – maybe I missed it but what is MAP criteria
> Map criteria is an analysis to determine the number of factors.
Results – Was there a cut off period? I mean, I know you enrolled owners of deceased dogs, but was there a time limit?
> Limit is determined by the owner and the association, but there is no specific limit at this time.
Line 139 – please explain why 98 out of a possible 101 were chosen.
> Three of the 101 questions were excluded because these questions were not appropriate for factor analysis. Specifically, factor loadings of 0.4 or less were excluded. And in this connection, we found the mistyping in the Table 1., and then we corrected them in Factors 13 and 14 in it.
Discussion:
Lines 193-4 and 224 appear to say similar things – low levels of aggression towards other dogs.
> These are similar but have different means. In lines 193-4, the dog's fear and anxiety toward the dogs is shown, and in line 224, the dog's aggression toward the dogs is shown.
Conclusion: Line 265 - you extracted factors that appear to indicate that dogs have the potential to become therapy dogs.
> We modified this sentence according to your advice on Line 266.